 

# The satiety hormone cholecystokinin gates reproduction in fish by controlling gonadotropin secretion

Lian Hollander-Cohen[1], Omer Cohen[1], Miriam Shulman[1], Tomer Aiznkot[1], Pierre Fontanaud[2,3], Omer Revah[4], Patrice Mollard[2,3], Matan Golan[1,5]*, Berta Levavi-Sivan[1]*

[1]Department of Animal Sciences, The Robert H. Smith Faculty of Agriculture, Food, and Environment, Hebrew University of Jerusalem, Rehovot, Israel; [2]Institute of Functional Genomics, University of Montpellier, Montpellier, France; [3]BioCampus Montpellier, University of Montpellier, Montpellier, France; [4]The Koret School of Veterinary Medicine, The Hebrew University of Jerusalem, Jerusalem, Israel; [5]Department of Poultry and Aquaculture, Institute of Animal Sciences, Agricultural Research Organization, Volcani Center, Rishon Letziyon, Israel

## eLife Assessment

This study presents **valuable** findings on the role of the satiety hormone cholecystokinin typically associated with feeding in the control of a pituitary hormone, FSH, which is a critical regulator of reproductive physiology. The authors provide **solid** pharmacological evidence that cholecystokinin is sufficient to regulate FSH and **compelling** genetic evidence that one of its receptors is required for gonadal development, with uncertainties remaining about the physiological regulation and necessity of the peptide. The work will be of interest to reproductive biologists, especially those with an interest in the endocrine control of fertility.

*For correspondence:
matan.golan@mail.huji.ac.il (MG);
berta.sivan@mail.huji.ac.il (BL-S)

Competing interest: The authors declare that no competing interests exist.

**Abstract** Life histories of oviparous species dictate high metabolic investment in the process of gonadal development leading to ovulation. In vertebrates, these two distinct processes are controlled by the gonadotropins follicle-stimulating hormone (FSH) and luteinizing hormone (LH), respectively. While it was suggested that a common secretagogue, gonadotropin-releasing hormone (GnRH), oversees both functions, the generation of loss-of-function fish challenged this view. Here, we reveal that the satiety hormone cholecystokinin (CCK) is the primary regulator of this axis in zebrafish. We found that FSH cells express a CCK receptor, and our findings demonstrate that mutating this receptor results in a severe hindrance to ovarian development. Additionally, it causes a complete shutdown of both gonadotropins secretion. Using in-vivo and ex-vivo calcium imaging of gonadotrophs, we show that GnRH predominantly activates LH cells, whereas FSH cells respond to CCK stimulation, designating CCK as the bona fide FSH secretagogue. These findings indicate that the control of gametogenesis in fish was placed under different neural circuits, that are gated by CCK.

## Introduction

In vertebrates, the processes of folliculogenesis, ovulation, and spermatogenesis are controlled by two gonadotropin hormones (GtHs), FSH and LH. In fish, LH and FSH are secreted by distinct cell

types in the pituitary. In recent years, different loss-of-function (LOF) studies revealed the stereo-typed function of each gonadotropin: FSH signaling controls folliculogenesis, whereas the role of LH is restricted to the induction of ovulation (*Takahashi et al., 2016*; *Zhang et al., 2015a*; *Chu et al., 2014*). According to the existing dogma, the secretion of both GtHs by gonadotrophs of the anterior pituitary gland is controlled by the hypothalamic neuropeptide GnRH, which is produced by a small population of neurons in the preoptic area (*Belchetz et al., 1978*). Studies conducted in mammals have shown that the differential control over gonadotropin secretion is attained via changes in frequencies and amplitude of GnRH pulses (*Savoy-Moore and Swartz, 1987*; *Stamatiades and Kaiser, 2018*; *Thompson and Kaiser, 2014*), as well as by a variety of other endocrine and paracrine factors that dictate whether the cells will secrete LH or FSH. In fish, the TGF-β family members activin, inhibin, and follistatin, as well as PACAP signalling have been shown to exert a differential effect on FSH and LH synthesis (*Yuen and Ge, 2004*; *Aroua et al., 2012*; *Yaron et al., 2001*; *Lin and Ge, 2009*). However, the hypothalamic mechanisms governing the differential release of FSH or LH in non-mammalian vertebrates remain largely unknown.

As in mammals, GnRH is considered the master regulator of gonadotropin secretion in fish. However, in recent years, its status as the sole neuropeptide regulating GtH secretion has been called into question, as other hypothalamic neuropeptides were shown to bypass GnRH and directly regu-late gonadotropin secretion (*Biran et al., 2008*; *Tsutsui et al., 2012*; *Biran et al., 2014a*; *Biran et al., 2014b*; *Ogawa et al., 2016*; *Cohen et al., 2020*; *Mitchell et al., 2020*; *Rajeswari and Unniappan, 2020*). Due to genome duplication events, fish brains express up to three forms of GnRH, of which one form (usually GnRH1) innervates the pituitary gland (*Tello et al., 2008*; *Okubo and Nagahama, 2008*). In some species, such as the zebrafish, that express only two forms of GnRH (GnRH2 and GnRH3), the gene encoding GnRH1 has been lost, and GnRH3 has become the dominant hypophysiotropic form (*Okubo and Nagahama, 2008*). For a yet unknown reason, in the zebrafish, even a complete absence of GnRH does not impair ovulation, as adult zebrafish with LOF of GnRH signaling are fertile (*Tanaka et al., 2022*; *Spicer et al., 2016*; *Whitlock et al., 2019*), suggesting that GnRH activity is either replaced by a compensation mechanism or it is dispensable for the control of gonadotropin release overall. In other species, such as medaka, the effects of GnRH are limited to the control of final oocyte maturation and ovulation via LH secretion (*Takahashi et al., 2016*). Since in both species the loss of GnRH does not affect gonadal development, the hypothalamic factor controlling FSH secretion in fish remains unknown (*Takahashi et al., 2016*; *Spicer et al., 2016*; *Marvel et al., 2018*).

Here, we addressed the question of the hypothalamic control of LH and FSH secretion in zebrafish. By mutating a previously identified CCK receptor highly expressed in FSH cells (*Hollander-Cohen et al., 2021*), we prove that CCK controls zebrafish reproduction by gating gonadotropin secretion. Using in vivo and ex vivo calcium imaging in zebrafish gonadotrophs to identify LH- and FSH-specific secretagogues, we show that the two types of gonadotrophs vary significantly in their activity patterns and that GnRH controls LH cells whereas FSH cells are preferentially activated by the satiety hormone CCK, which is also produced in the fish hypothalamus (*Himick and Peter, 1994*; *Sobrido-Cameán et al., 2020*) and its receptor is highly expressed in FSH cells (*Hollander-Cohen et al., 2021*). The results identify CCK as a novel crucial regulator of the reproductive axis and establish a neuroendo-crine link between nutritional status and reproduction in fish.

## Results

### CCK and its receptor are vital regulators of the HPG axis

While in mammals, both LH and FSH are secreted by the same cell population, in fish these gonado-tropins are produced by discrete cell types (*Hollander-Cohen et al., 2021*; *Golan et al., 2016*). We took advantage of this unique feature to search for the mechanism that regulates the differential secretion of LH and FSH in fish. We have previously reported (*Hollander-Cohen et al., 2021*) that FSH cells differ from LH cells by the expression of an FSH-specific type of CCK receptor. While three types of CCK receptors (CCKRs; CCKAR, CCKBR, and CCKBR(A)) are reported in the genome of fish, only one type (CCKBR(A), XM_017357750.2) is expressed in the pituitary gland (*Hollander-Cohen et al., 2021*; *Figure 1—figure supplement 1F*). In tilapia, the expression of this receptor is ~100- fold higher in FSH than in LH cells (*Figure 1A*). Nevertheless, LH cells also express the CCK receptor (CCKBR(A)) albeit at a lower level (*Figure 1A*). We, therefore, first validated its expression on gonadotroph cells

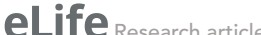

**Figure 1.** Effect of CCKBR(A) loss-of-function mutation on gonadal development. (**A**) Expression of CCKBR(A) and two identified gonadotropin-releasing hormone (GnRH) receptors in luteinizing hormone (LH) and Follicle-Stimulating Hormone (FSH) cells. Expression data were taken from a transcriptome of sorted pituitary cells of transgenic Nile tilapia *Oreochromis niloticus*, previously obtained by *Hollander-Cohen et al., 2021*. Each dot represents a FACS fraction from a bulk of 20 pituitaries (n=8, 4 groups of males and 4 groups of females). The expression of each gene in each cell type is normalised to its expression in non-gonadotroph pituitary cells. (two-way ANOVA, * p<0.05,). (**B**) RNA expression of CCKBR(A) (white) identified

*Figure 1 continued on next page*

*Figure 1 continued*

by hybridization chain reaction (HCR) in transgenic zebrafish pituitaries expressing RFP in LH cells (magenta) and GFP in FSH cells (green) (scale bar = 10 µm), the right panel is a magnification of the white square in the left panel. (**C**) Immunohistochemical staining of cholecystokinin (CCK) (white) in transgenic zebrafish expressing GFP in FSH cells (left panel, scale bar = 50 µm and 8 µm) or GFP in GnRH neurons (right panel, scale bar = 100 µm and 10 µm). In B and C, staining from adult fish was performed on whole head sections 15 µm thick. (**D**) H&E staining of body cross-sections (dorsoventral axis) of adult WT, heterozygous (CCKBR(A)$^{wt/+12}$, CCKBR(A)$^{wt/+7}$, CCKBR(A)$^{wt/-1}$), and KO zebrafish (CCKBR(A)$^{+12/+12}$, CCKBR(A)$^{+7/+7}$,CCKBR(A)$^{-1/-1}$). An inset of the red square in each image on the right displays a magnified view of the gonad. On the top right of each panel is the sex distribution for each genotype. (**E**) Gonad areas of mutant zebrafish. (n$_{((+/+),(+/-), (-/-))}$=10/6/17,one way ANOVA,, ****p<0.0001). (**F**) The distribution of cell types in the gonads of WT, heterozygous, and KO zebrafish. (n$_{((+/+),(+/-), (-/-))}$=10/6/17, two-way ANOVA, *p<0.05, **p<0.001, ***p<0.0001,****p<0.00001). (**G**) Gonadotroph mRNA expression in the pituitaries of the three genotypes (n$_{((wt/wt),(wt/-), (-/-))}$=10/8/9, one-way ANOVA, *p<0.05, **p<0.01, ***p<0.001).

The online version of this article includes the following source data and figure supplement(s) for figure 1:

**Source data 1.** Single RNA Guides used for CCKR KO with CRISPR-Cas9 and primers used for real-time polymerase chain reaction (PCR) of tissue distribution in tilapia.

**Figure supplement 1.** Supporting evidence for CCKBRA activity, expression and loss of function in the CRISPR mutants.

in zebrafish using in situ hybridization and found that the receptor is predominantly expressed in FSH cells (*Figure 1B*). To identify the source of CCK inputs we used immunohistochemistry to label CCK-expressing cells using a specific antibody against the eight amino acids of the mature CCK peptide. We found hypophysiotropic CCK-secreting neuronal projections innervating the pituitary near FSH cells and adjacent to GnRH axons (*Figure 1C*), indicating hypothalamic input of CCK into the pituitary gland.

To functionally validate the importance of CCK signaling, we used CRISPR-cas9 to generate LOF mutations in the pituitary-specific CCK receptor gene (CCKBR(A)). Three different mutations were induced by guide RNAs designed to target the fourth transmembrane domain of the protein, thus affecting the binding site of the receptor to its ligand (*Figure 1—figure supplement 1A–D*). Three mutations were identified to generate a LOF: insertion of 12 nucleotides (CCKBR(A)$^{+12}$), insertion of seven nucleotides (CCKBR(A)$^{+7}$), and deletion of one nucleotide (CCKBR(A)$^{-1}$). Modifications to the CCKBR(A) structure had significant implications for its function and ligand-binding capabilities, resulting in receptor inactivation (*Figure 1—figure supplement 1E*). An insertion of +12 nucleotides is particularly interesting because a four-amino acid insertion does not cause a frame shift in the coding sequence. However, this insertion of a four-amino acid sequence (Asp-Asp-Ser-His) at the start of TM4 of the CCKBR(A) receptor, specifically between amino acids 188 and 192, led to receptor inactivation. A study by *Foucaud et al., 2008* demonstrated that amino acid 192 is part of the binding site for the mammalian CCKR2. Our findings indicate that modifications in this region, significantly impact the receptor's ligand-binding characteristics and overall functionality (*Figure 1—figure supplement 1E*). Analysis of the phenotype of F2 adult fish (5–6 mo of age) revealed that while non-edited (wt/wt, n=15) and heterozygous fish (CCKBR(A)$^{wt/+12}$, CCKBRB$^{wt/+7}$, CCKBR(A)$^{wt/-1}$, n=20) displayed typical sex ratios and functional adult gonads, all homozygous fish (CCKBR(A)$^{+12/+12}$, CCKBR(A)$^{+7/+7}$,CCKBR(A)$^{-1/-1}$, n=12) were males with significantly small gonads (*Figure 1D–E*; mean gonad area KO = 3.8 ± 0.48 mm^2, Heterozygous = 23.52 ± 10.6, WT = 78 ± 7.4 mm^2). The testes of mutant males displayed an immature phenotype as they were populated mostly by early stages of testicular germ cells (mostly spermatogonia and spermatocytes) and contained low volumes of mature spermatozoa compared to their WT and heterozygous siblings (*Figure 1F*). Heterozygous fish were also affected and displayed significantly lower amounts of spermatozoa compared to the WT fish. Interestingly, the CCKBR(A) LOF fish do not phenocopy zebrafish with a loss of FSH (*Zhang et al., 2015b*; *Chu et al., 2015*). Instead, the phenotype of the CCKBR(A) LOF closely resembles the condition reported for zebrafish that have LOF mutations in both gonadotropin genes (*Zhang et al., 2015b*; *Chu et al., 2015*). Indeed, our mutants show decreased expression levels of both *lhβ* and *fshβ* genes (*Figure 1G*), suggesting that loss of CCK signaling affects both LH and FSH. Tissue distribution of the CCK receptors reveals the high expression of CCKBR(A) in the pituitary and in the hypothalamus (*Figure 1—figure supplement 1F*), which are both central areas of the HPG axis. CCK expression in the brain is directly connected to the metabolic status of the fish, as starved fish have a significant decrease in CCKB expression compared to fed fish (*Figure 1—figure supplement 1G*). We next sought to identify the exact effect of the two major HPG regulators GnRH and CCK on the activity of LH and FSH cells using calcium imaging.

## LH and FSH cells exhibit distinct calcium activity in vivo

The unique segregation of LH and FSH cells in fish provides an opportunity to identify genes and pathways that specifically regulate each gonadotroph. To that end, we generated transgenic zebrafish in which both LH- and FSH-producing cells express the red genetically-encoded calcium indicator RCaMP2 (*Inoue et al., 2015*), whereas FSH cells also express GFP (*Tg(FSH:RCaMP2, LH:RCaMP2, FSH:GFP)*; *Figure 2A*). These fish allow simultaneous monitoring of calcium activity in LH and FSH cells as a readout for cell activation, while distinguishing between the two cell types.

To follow the activity of LH and FSH gonadotrophs in live zebrafish, we developed a novel preparation for imaging the pituitary gland at single-cell resolution while maintaining the in vivo context. In this preparation, the pituitary gland was exposed from its ventral side. The immobilized zebrafish were placed under a two-photon microscope with a constant flow of water over the gills (*Figure 2B*), ensuring sufficient oxygen supply to the gills and blood flow to the gland (*Figure 2— video 1*).

Imaging of the gonadotrophs in vivo revealed distinct types of calcium activity in the two cell types (*Figure 2C-E*, *Figure 2—figure supplement 1A*). In the basal state, LH cells were mostly silent, whereas FSH cells exhibited short (mean of 10.08 s half width) calcium bursts (mean of 0.6 ΔF/F; *Figure 2C* (fish1) and *Figure 2D*; *Figure 2—video 2*). These calcium events were sparse, i.e. between 1 and 7 transients in each cell in 10 min, and disorganized, as max cross-correlation coefficient values ranged from 0.3 to 0.9 (*Figure 2E* fish1). In seven out of ten zebrafish, we observed an event in which LH cells exhibited a single long (mean of 75.55 s half-width) and strong (mean of 1.5 ΔF/F) calcium rise (*Figure 2C* (fish2) and D). This event was synchronized between LH cells (mean max cross-correlation coefficient values of 0.89 ± 0.003) and, in three out of seven fish, it was followed by a less synchronized (mean max cross-correlation coefficient values, 0.66 ± 0.011,) calcium rise in FSH cells (*Figure 2C* (fish2) and *Figure 2D*, *Figure 2—video 3*). On average, the max cross-correlation coefficient values of all active LH cells were significantly higher compared to FSH cells (*Figure 2F*, *Figure 2—figure supplement 1B*), which reflects the strong gap-junction mediated coupling between LH cells that does not exist in FSH cells (*Golan et al., 2016*). We did not observe significant sexual dimorphism in correlation value distribution in either LH or FSH cells (five males and five females). Since most FSH calcium transients were not associated with a rise in calcium in LH cells, we speculated that a cell type-specific regulatory mechanism drives the activity of the two cell types and GnRH and CCK are the primary candidates.

## GnRH primarily activates LH cells

Next, we sought to characterize the response of the cells to GnRH, their putative common secret-agogue. To determine the effect of GnRH on gonadotroph activity, we utilized an ex vivo preparation that preserves the brain-pituitary connection intact (*Figure 3A*). Without stimulation, LH cells were either silent or exhibited small and short calcium transients (2–8.8 s half-width, amplitude 0.23–0.66 ΔF/F; *Figure 3B*; *Figure 3—video 1* (basal); *Figure 3—figure supplement 2*) which were synchronized between small groups of neighbouring cells (2–14 cells per fish with a max cross-correlation coefficient >0.5; *Figure 3—figure supplement 1*). Independent of the basal activity of LH cells, in 80% of the fish, FSH cells elicited short and intense calcium bursts that had no clear organization (5.8– 12.13 s half-width, amplitude 0.63–1.03 ΔF/F; *Figure 3B*, *Figure 3—figure supplement 2*; *Figure 3— video 2*).

For stimulation, we applied a GnRH3 analog that effectively binds to the three types of GnRH receptors that exist in fish and is also commercially used for spawning induction in fish (*Zohar and Mylonas, 2001*). In response to GnRH puff application (300 μl of 30 μg/μl, 78.74 μM), LH cells exhibited a strong and slow calcium rise (average half width 48.6 s and average amplitude of 1.99 ΔF/F; *Figure 3C*, *Figure 3—figure supplement 2*; *Figure 3—video 1* (GnRH) and 3), which was synchronized between the cells, as observed by the increase in correlation values from 0.26 ± 0.02–0.66 ± 0.05 (*Figure 3D*). In contrast, in FSH cells only 50% of the fish displayed an increase in cross-correlation values from the basal state (*Figure 3D*; *Figure 4—figure supplement 1A*). Overall, whereas GnRH elicited a response in 95% of LH cells, only 56% of FSH cells responded to the treatment in the same fish (*Figure 3E*). Due to this inconsistent response of FSH cells to GnRH stimuli, we speculated that CCK might regulate their activity.



**Figure 2.** Luteinizing hormone (LH) and follicle-stimulating hormone (FSH) cells exhibit distinct spontaneous activity patterns in vivo. (**A**) A confocal image of the pituitary shows RCaMP2 expression in both cell types and GFP expression in FSH cells (scale bar = 100 μm). (**B**) A diagram describing the setup of the in vivo experiments. The dissected zebrafish were placed in a chamber with a constant flow of water to the gills and imaged in an upright two-photon microscopy. (**C**) A representative image of in vivo calcium activity (see *Figure 2—video 3*). On the top left is a merged image depicting FSH cells in green and LH cells in magenta. The other top panels show sequential calcium imaging calcium rise in FSH cells is marked by white arrows. The bottom panels show the calcium of LH and FSH cells in two different imaged pituitaries, one where only FSH cells were active (Fish 1) and another where both cell types were active (Fish 2, traces ΔF/F, see *Figure 2—figure supplement 1A*. for heatmap of the calcium traces, scale bar = 20 μm). (**D**) The properties of spontaneous calcium transients (ΔF/F) in LH cells and FSH cells in three males and one female and their means. (unpaired *t*-test, **p<0.001). Analysis was performed using pCLAMP 11. (**E**) Left: cross-correlation analysis between active ROI to the rest of the cell. The color-coded data points are superimposed on the imaged cells and represent the maximum cross-correlation coefficient between a calcium trace of a region of interest (ROI) and that of the rest of the cells in the same population. Right: is a matrix of maximum cross-correlation coefficient values between all the cells. (scale bar = 20 μm). (**F**) Summary of the mean max cross-correlation coefficient values of calcium traces in each cell population of repeated in vivo calcium imaging assays (n $_{(calcium\ sessions)}$=16, see *Figure 2—figure supplement 1B* for all measurements,. unpaired *t*-test, ***p<0.0001).

The online version of this article includes the following video and figure supplement(s) for figure 2:

**Figure supplement 1.** Calcium activity and cross correlation of FSH and LH cells In vivo.

**Figure 2—video 1.** Blood flow in the pituitary of live fish.

https://elifesciences.org/articles/96344/figures#fig2video1

**Figure 2—video 2.** Calcium imaging of luteinizing hormone (LH) and follicle-stimulating hormone (FSH) cells in live fish where only FSH cells were active.

https://elifesciences.org/articles/96344/figures#fig2video2

*Figure 2 continued on next page*

*Figure 2 continued*

**Figure 2—video 3.** Calcium imaging of luteinizing hormone (LH) and follicle-stimulating hormone (FSH) cells in live fish where both cell types (LH and FSH) were active.

https://elifesciences.org/articles/96344/figures#fig2video3

## Cholecystokinin directly activates FSH cells

To functionally test the effect of CCK on gonadotroph activity, we applied the peptide (300 µl of 30 µg/ml CCK, 78.74 µM) to our ex vivo preparation and monitored the calcium response of the cells. CCK elicited a strong calcium response (40.3–172- s half-width, mean amplitude of 1.44 ΔF/F) in FSH cells, while in some of the fish, a lower response was observed in LH cells (*Figure 4A and B*; *Figure 4—figure supplement 1B*; *Figure 4—video 1*). In all analyzed fish (n=7), all FSH cells responded to CCK, whereas the number of LH cells that responded to the stimulation varied widely (20–100%, n=7; *Figure 4C*). The calcium response in FSH cells was highly synchronized (mean max cross-correlation coefficient, 0.7 ± 0.04; *Figure 4B*; *Figure 4—figure supplement 1B*). By contrast, the response of LH cells to CCK application was characterized by low mean of max cross-correlation coefficient values (0.43 ± 0.05; *Figure 4B*; *Figure 4—figure supplement 1B*). These results indicate that in fish, CCK preferentially activates calcium rise in FSH cells, albeit with a weaker activation of LH cells.

## Differential calcium response underlies differential hormone secretion

The calcium response observed in LH and FSH cells upon GnRH and CCK stimulation indicates a preferential stimulatory effect of the neuropeptides on each cell type. The effect of these neuropeptides on LH and FSH secretion was examined in order to determine the functional outcome of the stimulation. For that, we collected the medium perfused through our ex vivo system (*Figure 3A and B*) during our calcium imaging assays. The medium was subsequently used to measure LH and FSH secretion using a specific ELISA validated for zebrafish GTHs (*Hollander-Cohen et al., 2018*). As expected, the calcium response to GnRH in LH cells was followed by a significant rise in LH secretion (n=5; *Figure 5A*, LH concentration ranging from 0.1 to 2 ng/ml). In contrast, FSH cells responded with a very low rise in hormonal secretion in response to GnRH treatment (FSH concentration ranging from 0.003 to 0.06 ng/ml) that was not significant from the basal secretion. Conversely, the application of CCK elicited a significant calcium rise in FSH cells followed by an elevation of FSH concentration in the medium (FSH concentration ranging from 0.5 to 5 ng/ml), whereas in LH cells, no significant effect was observed on calcium activity, and the slight increase in LH secretion was not significant (n=5; LH concentration ranging from 0.25 to 0.75 ng/ml *Figure 5B*). These results were reproduced in vivo, as CCK injection significantly increased the expression and secretion of FSH (*Figure 5C and D*) whereas the response to GnRH did not reach statistical significance (p=0.069). GnRH only affected LH expression in the pituitary (*Figure 5—figure supplement 1*).

Taken together, these results suggest that GnRH and CCK preferentially activate calcium-dependent secretion in LH and FSH cells, respectively, and induce the release of these gonadotropins from the pituitary gland.

## Discussion

GnRH has long been considered the common stimulator of gonadotropin secretion in vertebrates. However, accumulating evidence for GnRH-independent FSH secretion in several mammalian species has questioned the regulatory role of GnRH (*McCann et al., 2001*; *Padmanabhan et al., 1997*; *Culler and Negro-Vilar, 1987*; *Pau et al., 1991*). Moreover, in fish, normal ovarian development in hypophysiotropic GnRH loss-of-function mutants (*Takahashi et al., 2016*; *Marvel et al., 2018*), together with the lack of FSH cells response to GnRH stimuli in pituitary cell culture (*Lin and Ge, 2009*), further highlights the existence of an unknown FSH regulator other than GnRH. Here, we reveal that in zebrafish, CCK, a satiety hormone, gates reproduction by directly regulating GtH secretion. We show that GnRH preferentially controls LH secretion and identify CCK as the long-sought hypothalamic FSH secretagogue. Our results indicate that while fish gonadotrophs were segregated into two different populations and placed under the control of two distinct neuropeptides during evolution, a common hypothalamic pathway gates the secretion of both gonadotropins. Interestingly, in contrast to GnRH, the novel CCK regulation identified under the current study has a more substantial effect on the



**Figure 3.** Gonadotropin-releasing hormone (GnRH) induces a synchronized increase of calcium in all luteinizing hormone (LH) cells and partial increase in follicle-stimulating hormone (FSH) cells. (**A**) Top: Image of a dissected head with pituitary exposed from the ventral side of the fish used for the ex vivo assays (OB, olfactory bulb; OC, optic chiasm; PIT, pituitary; TH, thalamus; MO, medulla oblongata). Bottom: A diagram describing the ex vivo setup with a constant flow of artificial cerebrospinal fluid (ACSF), a side tube to inject stimuli, and a collecting tube. (**B**) A representative analysis of basal calcium activity of LH and FSH cells. The left panel is a heatmap of calcium traces (ΔF/F), where each line represents a cell, with the mean calcium trace on top, the separated line at the bottom of each heatmap is the calcium trace of the chosen region of interest (ROI). The color-coded data points on the right are superimposed on the imaged cells and represent the maximum cross-correlation coefficient between a calcium trace of an active chosen ROI to those of the rest of the cells in the same population, the matrix on the right represent the maximum cross-correlation coefficient values between all the cells (see *Figure 3—figure supplement 2* for additional cell activity parameters, scale bar = 20 μm). (**C**) An analysis of calcium response to GnRH stimulation in two representative imaging sessions, fish 1 where both LH and FSH cells respond, and fish 2 where only LH respond. (see *Figure 4—*

*Figure 3 continued on next page*

*Figure 3 continued*

*figure supplement 1* for detailed coefficient values distribution in each fish, scale bar = 20 μm). (**D**) The mean of max cross-correlation coefficient values in each cell type under each treatment (n=10, 3 males, 7 females, one-way ANOVA, ****p<0.0001, see *Figure 4—figure supplement 1A* for cell-specific vlaues). (**E**) The percentage of cells responsive to GnRH stimulus (i.e. coefficient values higher than the 80 percentiles of basal values). Each dot represents one fish (n=10, 3 males, 7 females, unpaired *t*-test, ***p<0.001).

The online version of this article includes the following video and figure supplement(s) for figure 3:

**Figure supplement 1.** Calcium analysis of basal activity in luteinizing hormone (LH) cells reveals the synchronized spontaneous activity of small clamps of cells.

**Figure supplement 2.** Calcium signal properties of LH and FSH cells.

**Figure 3—video 1.** Ex-vivo calcium imaging of luteinizing hormone (LH) cells basal activity and gonadotropin-releasing hormone (GnRH)-stimulated calcium wave of LH and follicle-stimulating hormone (FSH) cells, using confocal microscopy.

https://elifesciences.org/articles/96344/figures#fig3video1

**Figure 3—video 2.** Ex-vivo calcium imaging of follicle-stimulating hormone (FSH) cells basal activity, using confocal microscopy.

https://elifesciences.org/articles/96344/figures#fig3video2

**Figure 3—video 3.** Ex-vivo gonadotropin-releasing hormone (GnRH)-stimulated calcium wave of only luteinizing hormone (LH) cells; 2500 frames, 4 Hz image, 150 frames per second.

https://elifesciences.org/articles/96344/figures#fig3video3

gonadotrophic axis, as revealed in our mutants, while the disruption of GnRH and its receptors didn't lead to any drastic effect on reproduction (*Tanaka et al., 2022*; *Spicer et al., 2016*).

The role of CCK as a satiety hormone has been demonstrated in multiple species of mammals and fish (*Lõhmus et al., 2008*; *Zhang et al., 2017*; *Lo et al., 2010*). From an evolutionary perspective, allocating a satiety hormone for gating reproduction serves the unique demands of the life history of oviparous species. Egg-laying and placental animals display a marked difference in their reproductive energy allocation strategies. In oviparous species, the pre-ovulatory processes of gonadal development involving vitellogenin synthesis and deposition into the developing oocyte, also known as folliculogenesis, constitute the main nutritional challenge during the female reproductive cycle. Thus, in oviparous species, folliculogenesis is gated by the animal's nutritional status. Since folliculogenesis and spermatogenesis are controlled by gonadotropin signaling from the pituitary, nutritional gating of gonadotrophs, may serve as an effective pathway to inhibit gonad development under limiting energetic balance. Our data suggest that, indeed, dedicated hypothalamic neurons have developed to integrate metabolic cues, such as food abundance and somatic condition, to induce the energetically costly process of reproduction in fish.

Since CCK is a regulator of satiety in fish (*Himick and Peter, 1994*; *Zhang et al., 2017*; *Volkoff, 2006*), this hypothalamic circuit directly links the metabolic status of the fish to its reproductive capacity (*Figure 6*). Considering the recent report of an FSH-regulating role for CCK in the distantly related species medaka (*Uehara et al., 2023*), our findings represent a highly conserved mechanism for controlling reproduction in fish.

In the CCKBR(A) LOF mutants, gonad development was disrupted and led to an all-male population with underdeveloped testes. Female gonad development in fish is directly linked to FSH signalling activity, as shown by genetic mutation of FSH receptors that leads to female gonad arrest and differentiation into male gonads as the fish mature (*Zhang et al., 2015a*). The lack of females in our LOF mutant, together with the male's infertility, suggests a direct disruption of both LH and FSH circuitry. We further show that CCK directly controls FSH cells by innervating the pituitary gland and binding to specific receptors that are particularly abundant in FSH gonadotrophs. However, our calcium imaging results and the LOF mutants demonstrate that CCK also activates LH cells to some extent. This activation may either be direct, as LH cells were also shown to express the CCK receptor (*Hollander-Cohen et al., 2021*) albeit at a lower level, or indirectly, by affecting LH cells via activation of GnRH or other neurons. The latter pathway can be wired through the close apposition of GnRH3 and CCK terminals in the zebrafish pituitary, as shown in this work. Additional support is the evidence in mammals where GnRH neurons express CCK receptors and CCK directly affects the migratory pathway of GnRH during development (*Giacobini et al., 2004*; *Giacobini and Wray, 2007*).

Our identification of CCK-producing axons innervating the zebrafish pituitary suggests a predominantly central CCK-dependent control of FSH release. While in mammals, hypothalamic regulation

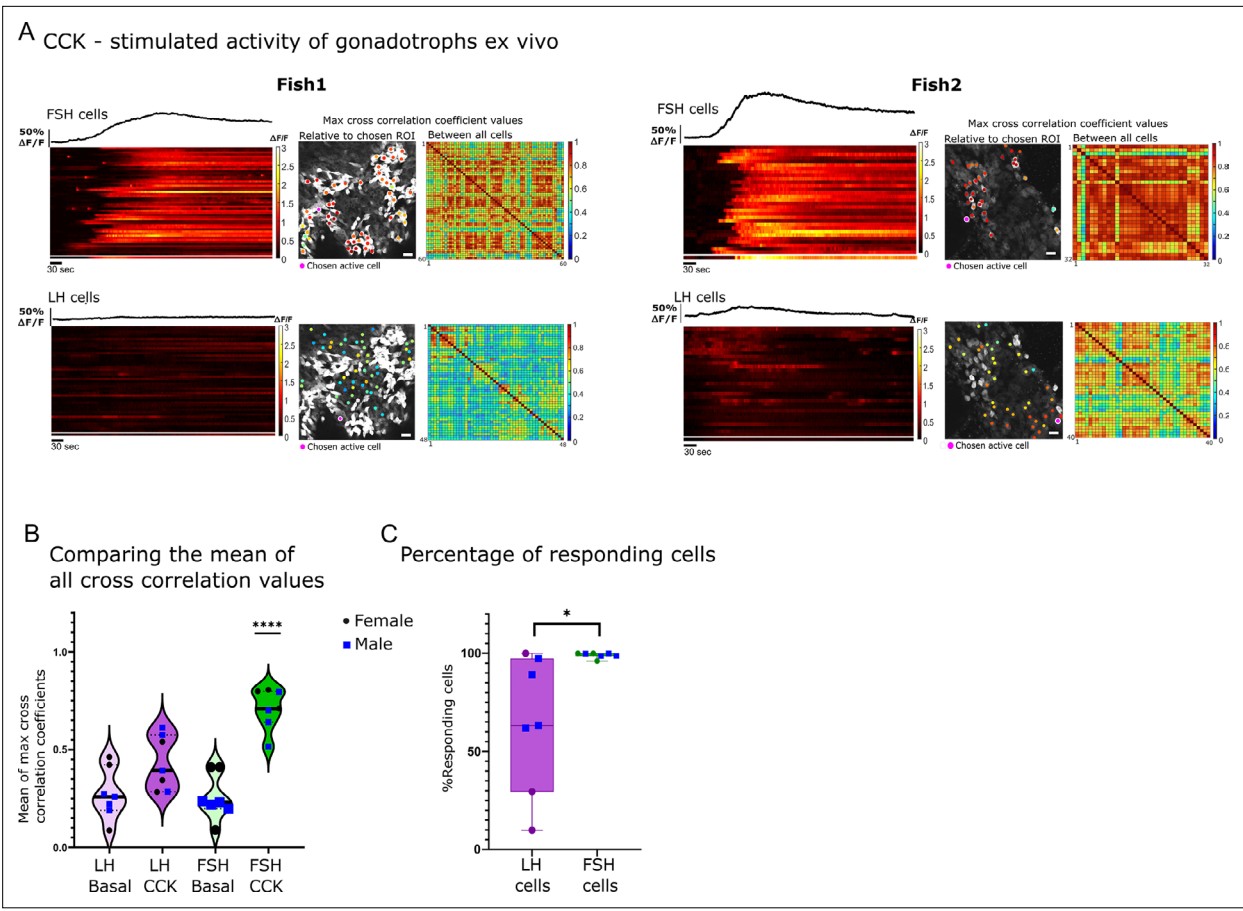

**Figure 4.** Follicle-stimulating hormone (FSH) cells are directly stimulated by cholecystokinin (CCK). (**A**) Example of calcium analysis of FSH and luteinizing hormone (LH) cells during CCK stimulation: fish1 with only FSH cells responding, and fish2 with FSH and LH cells responding. For each fish, the left panels are a heatmaps of calcium traces (ΔF/F), where each line represents a cell. On top of each heatmap is a graph showing the mean calcium trace, the separated line at the bottom of the heat map is the calcium trace of the chosen region of interest (ROI). On the right are color-coded data points that are superimposed on the imaged cells, showing the maximum cross-correlation coefficient between a calcium trace of a chosen active ROI and those of the rest of the cells in the same population, next to it is a matrix of max cross-correlation coefficients between all the cells. (scale bar = 20 μm) (**B**) The mean of max cross-correlation coefficient values in each cell type (n=7, 4 males, 3 females, one-way ANOVA, ****p<0.0001, see *Figure 4—figure supplement 1B* for detailed coefficient values distribution in each fish). (**C**) The percentage of active cells (i.e. a coefficient value higher than the 80 percentiles of basal levels) during CCK stimulation. Each dot represents one fish (n=7, 4 males, 3 females, unpaired *t*-test, *p<0.05).

The online version of this article includes the following video and figure supplement(s) for figure 4:

**Figure supplement 1.** Max cross-correlation coefficient values between the calcium activity of all measured luteinizing hormone (LH) (magenta) or follicle-stimulating hormone (FSH) (green) cells in each fish.

**Figure 4—video 1.** Ex-vivo calcium imaging of luteinizing hormone (LH) and follicle-stimulating hormone (FSH) cells stimulated with cholecystokinin (CCK), using confocal microscopy.
https://elifesciences.org/articles/96344/figures#fig4video1

of the pituitary is gated by the median eminence, in fish, hypothalamic neurons directly innervate the pituitary and secrete their regulatory peptide close to the destination cells (*Golan et al., 2015*). Similar innervation of CCK was observed in ancient jawless fish, such as lamprey (*Sobrido-Cameán et al., 2020*), and modern teleosts, such as the goldfish (*Himick and Peter, 1994*). Modern oviparous tetrapods were also shown to express the CCK receptor in their pituitary glands (*Wan et al., 2023*), indicating that this regulatory circuit may be evolutionarily conserved and common to oviparous vertebrates. However, since CCK is produced in the gut as well as in the central nervous system, we cannot rule out circulating CCK as a possible activator of GtH cells. Nevertheless, the direct innervation of CCK terminals into the pituitary gland and the concomitant increase of CCK in the gut and in the brain in response to feeding (*Zhang et al., 2017*; *Gomes et al., 2022*; *D'Agostino et al., 2016*)



**Figure 5.** The stimulated calcium activity of luteinizing hormone (LH) and follicle-stimulating hormone (FSH) cells is associated with hormone secretion. (**A and B**) Top: Graphs showing the mean calcium trace of 10 LH cells (left panel) or FSH cells (right panel) from consecutive imaging sessions before, during, and after the application of the stimulus (gonadotropin-releasing hormone, GnRH or cholecystokinin, CCK). Bottom: Secretion of LH or FSH before

*Figure 5 continued on next page*

*Figure 5 continued*

or after GnRH (**A**) or CCK (**B**) stimulation (dots from the same imaged pituitaries are connected with a line; n=5, paired t-test, *p<0.05). (**C**) FSH mRNA transcription in the pituitary 2 hr after injection of CCK or GnRH into live fish (n=10, 5 females and 5 males, see *Figure 5—figure supplement 1* for LH expression; one-way ANOVA, *p<0.05, **p<0.01). (**D**) FSH plasma levels after CCK or GnRH injection into live fish (n=10, 5 females and 5 males), (one-way ANOVA, *p<0.05, ***p<0.001).

The online version of this article includes the following source data and figure supplement(s) for figure 5:

**Source data 1.** The primers used for real-time polymerase chain reaction (PCR) of pituitary gene expression in the in vivo assay (*Figure 5*).

**Figure supplement 1.** Luteinizing hormone (LH) expression in the pituitary after in vivo injection of different concentrations of cholecystokinin (CCK) and gonadotropin-releasing hormone (GnRH).

suggest that the two sources of CCK are interconnected. In this context, the vagal nerve may serve as a possible gut-brain communication route, as it was shown to relay satiety signals to the hypothalamus via CCK (*Borgmann et al., 2021*; *Clemmensen et al., 2017*), thus forming a parasympathetic regulatory loop onto the hypothalamo-pituitary-gonadal axis.

Unlike the situation in fish, CCK LOF mice can reproduce (*Lo et al., 2010*), reflecting that the main metabolic challenge in the reproductive cycle of mammals is controlled by placental gonadotropins rather than by the hypothalamic-pituitary axis. The identified functional overlap in the hypothalamic control of both gonadotropins in zebrafish efficiently serves to gate reproduction by a single neuropeptide. Importantly, similar functional overlap also exists in the potency of GnRH to activate FSH cells, corresponding to a previous finding where LH and FSH are co-secreted during the female spawning cycle (*So et al., 2005*). However, since FSH cells express a different type of GnRH receptor (*Hollander-Cohen et al., 2021*), their activation is less consistent and results in reduced gonadotropin secretion. Moreover, in the zebrafish, as well as in other species, the functional overlap in gonadotropin signaling pathways is not limited to the pituitary but is also present in the gonad, through the promiscuity of the two gonadotropin receptors (*So et al., 2005*; *Hollander-Cohen et al., 2019*). This multilevel overlap creates functional redundancy that grants the reproductive system a high level of robustness and ensures the species' persistence.

Overall, our findings propose an updated view of the regulation of gonadal function in fish, in which metabolically driven hypothalamic circuits evolved to control gonad development. In addition to the novel insight into the evolution and function of the reproductive axis in oviparous animals, these findings are also of particular importance in the context of aquaculture, which has become the dominant supplier of fish for human consumption in the face of declining yields from wild fisheries (*Froehlich et al., 2018*; *Anderson et al., 2017*). With the identification of the CCK circuit as a regulator of folliculogenesis and the main gateway between metabolic state and reproduction, novel tools targeting this pathway can now be designed to manipulate gonadal development and overcome challenges in gamete production and the control of puberty onset in farmed fish (*Zohar, 1989*; *Taranger et al., 2010*).

## Materials and methods
### Sample size and replication
Sample size varied between experiments, depending on the number of transgenic fish allocated for each experiment. The minimum sample size was three. Each data set describes biological replicates.

### Data inclusion/exclusion criteria
Fish were excluded from analysis if they showed no calcium activity or if tissue movements during the full recording session prevented reliable calcium analysis. Data or samples were not excluded from analysis for other reasons.

**Figure 6.** A model summarizing the two suggested regulatory axes controlling fish reproduction. The satiety-regulated cholecystokinin (CCK) neurons activate follicle-stimulating hormone (FSH) cells. luteinizing hormone (LH) cells are directly regulated by gonadotropin-releasing hormone GnRH neurons that are gated by CCK, photoperiod, temperature, and behaviour, eventually leading to final maturation and ovulation. Bottom image schematically represents the relative timescale of the two processes and the associated gonadotropin levels. Created with BioRender.com/q88k105.

### Randomization

Fish used for the experiments were randomly selected and randomly assigned to experimental groups. All GtH cells that could be detected in the selected fish were used for the analysis and thus, there was no requirement for randomization of cell selection.

## Animals

All experiments were approved by the Animal Welfare and Ethical Review Body of Languedoc-Roussillon (APAFIS#745–2015060114396791) and by the Experimentation Ethics Committee of the Hebrew University of Jerusalem (research number: AG-17–15126, Date: April 30, 2017). Zebrafish were housed according to standard conditions. Fertilized eggs were incubated at 28.5 °C in E3 medium. For the current study, the transgenic line *tg(LH:RCaMP2,FSH:RCaMP2)* was generated by co-injection of two constructs (*FSH:RCaMP* and *LH:RCaMP*) to embryos, the positive F1 fish expressing RCaMP2 in both cell types were crossed again with *tg(FSH:GFP)* to generate the triple transgenic fish *tg(FSH:R-CaMP2, FSH:GFP, LH:RCaMP2)*. Other transgenic lines used were *tg(FSH:GFP)*, and *tg(GnRH:GFP)* and *tg(LH:RFP, FSH:GFP)* (*Golan et al., 2014*). All lines and construct are available upon request from the authors.

## Plasmid construction

All expression plasmids were generated using the Tol2kit (*Kwan et al., 2007*) and Gateway system (Invitrogen). Briefly, entry clones were generated by the addition of appropriate adaptors to DNA fragments via polymerase chain reaction (PCR) amplification. Amplicons were then recombined into donor vectors using BP recombination. A 5′-entry clone (p5E), a middle entry clone (pME), and a 3′-entry clone (p3E) were then recombined through an LR reaction into an expression vector carrying tol2-recognition sequences and either an mCherry or a GFP heart marker (pDestTol2CG). The LH and FSH promoters (*Golan et al., 2014*) were cloned from Nile tilapia genomic DNA and inserted into pDONRP4-P1R to generate p5′-LH and p5′-FSH; those clones had been previously shown to be effective in marking LH and FSH cells in zebrafish (*Golan et al., 2016*). R-CaMP2 (*Dana et al., 2016*) was a gift from H. Bito (University of Tokyo) and was cloned into pDONR221 to generate pME-R-CaMP2. For GFP expression, the middle clone pME-EGFP was used. The expression vectors containing p5E-**LH**:PME-R-CaMP2:p3E-PolyA (red heart marker) and p5E-**FSH**:PME-R-CaMP2:p3E-PolyA (green heart marker) were co-injected to create *tg(LH&FSH:CaMP2)* fish, which were later crossed with *tg(FSH:GFP)* fish to generate the triple transgenic fish.

## In vivo calcium imaging

Adult fish were anesthetized using 0.6 µM tricaine and immobilized by IP injection of 10 µl α-tubocurarine (5 mM; Sigma-Aldrich). To expose the pituitary, the jaw of the fish containing the dentary and part of the articular bones, together with a small slice from the mucosa overlying the palate, were removed by blunt dissection under a stereomicroscope, the total duration of the dissection was 10–15 min. The fish was then placed in a modified chamber, where the hyoid bone was gently pushed backward using a thin silver wire. A tube with a constant flow of fresh system water was placed in the jaw cavity in front of the gills. Heartbeat was monitored during calcium imaging as an indicator of viability. Each imaging session lasted 10 min and was repeated several times for each fish.

Calcium imaging was performed using a FVMPE RS two-photon microscope (Olympus) setup with an InSight X3 femtosecond-pulsed infrared laser (Spectra-Physics) and a 25x, numerical aperture 1.05 water-immersion objective (XLPLN25XWMP2, Olympus). The laser wavelength was tuned to 940 nm for GFP or 1040 nm for R-CaMP2. Calcium signals were recorded by time-lapse acquisition using galvanometric scanning mode and conventional raster scanning with a frequency of up to 10 Hz.

## Ex vivo calcium imaging

Adult fish were euthanized using ice-cold water and decapitated. The head was transferred to ice-cold ACSF (124 mM NaCl, 3 mM KCl, 2 mM CaCl$_2$, 2 mM MgSO$_4$, 1.25 mM NaH$_2$PO$_4$, 26 mM NaHCO$_3$, and 10 mM glucose (pH 7.2)) perfused with 5% CO$_2$ and 95% O$_2$. The heads were further dissected under a stereomicroscope. The ventral side of the head, including the jaw, gills, and mucus, was removed using fine forceps and microscissors, the optic nerves were cut and the eyes were removed. Next, the bone at the base of the skull that covers the pituitary (sella turcica) was removed using fine forceps. The head was placed in a dedicated chamber (*Figure 3B*) and stabilized with a slice anchor. The chamber had one inlet and one outlet, allowing for a constant flow of ACSF at a rate of 1 ml/min, and an additional inlet that was placed in proximity to the tissue for injections of stimuli. The total volume of ACSF in the chamber was 3 ml.

Imaging was performed in an inverted confocal fluorescent microscope (Leica SP8). R-CaMP2 activity was imaged at 4–10 Hz using the resonant scanner. The laser wavelength was 540 nm and the emission band was 630 nm. Images were taken using the 20x objective. Each imaging session lasted 10 min. Before and after each session, the tissue was imaged once for GFP (excitation, 488 nm, and emission, 530 nm) and R-CaMP2 (540 nm and 590 nm) in two separate channels. Images were processed using the Fiji program (*Schindelin et al., 2012*) for resolution using the Gaussian Blur filter and motion correction using the moco plugin (*Dubbs et al., 2016*).

To measure hormone secretion in response to salmon GnRH analogue (D-Ala6,Pro9-Net)-mammalian GnRH (Bachem Inc, Torrance, CA) or zebrafish CCK [D-Y-[SO3H]-L-G-W-M-D-F, synthesized by GL Biochem] stimulation, the medium was collected through the chamber outlet during the entire imaging session (10 ml per session in total). Each pituitary was imaged three times for 10 min: before, during, and after stimulation. The stimulus was applied manually as a pulse of 300 µl of 30 µg/ µl (78.74 µM) peptide during the first 30 s of the imaging session. Hormones were measured in the fractions before and after stimulation using ELISA developed for common carp, which was established in the Levavi-Sivan lab using recombinant carp gonadotropins produced in yeast, and had been previously shown to be suitable for zebrafish (*Whitlock et al., 2019*).

Tissue viability was validated by monitoring the morphology and activity of the cells, looking at granulation and calcium changes. This specific preparation was also viable 2 hr after dissection, when all the cells responded to the different stimuli.

## Analysis of Ca2+ imaging data

A composite of GFP-positive and RCaMP-positive cells was created to distinguish between LH and FSH cells. Regions of interest corresponding to each cell in the imaged plane were manually drawn using Fiji. Two separated ROI sets were created, FSH cells (GFP-positive) and LH cells (R-CaMP2-positive and GFP-negative). Using the Fiji ROI manager, two datasets were created: a data sheet containing the mean gray values in each frame during the complete image sequence, and a data sheet containing the ROI centroids. Sheets and images were then processed using MATLAB R2017a.

Traces were normalized using the equation $\Delta F = \frac{Ft}{F0} - 1$, where $F0$ is the lowest value in the means calculated from every N frame in the complete trace (N=sampling frequency×5). When the sampling rate was higher than 4 Hz, we applied a low pass filter with a cut-off frequency of 2/ (sampling frequency /2).

## Analysis of correlations coefficients between cells

Cross-correlation coefficients represent the maximum coefficient value between all cells or relative to a chosen ROI. From each set of traces, we obtained cross-correlation sequence ranges from -maxlag to maxlag, and the values were normalized such that autocorrelations at 0 lag equalled 1. For each set, we demonstrate only the maximum correlation values. The values are represented in a dot plot superimposed on the cells according to their centroid values when compared to the chosen ROI (pink dot), or as a heatmap when correlation is between all the cells. For violin plots, the mean of the maximum cross-correlation coefficient values in each fish was further visualized and analyzed for statistical significance using Prism 9 (GraphPad, San Diego, CA).

## HCR for CCKBR(A) and immunostaining of CCK

Staining was performed on whole head slices, as previously described (*Golan et al., 2015*). Briefly, whole heads were fixed overnight in 4% paraformaldehyde (PFA) and then decalcified for 4–7 d in 0.5 M EDTA at 4 °C. Subsequently, heads were cryoprotected in 30% (wt/vol) sucrose, frozen in an OCT embedding compound, and cryosectioned at a thickness of 15 µm. For immunostaining, head sections from transgenic zebrafish *tg(GnRH:GFP)* and *tg(FSH:GFP)* (*Golan et al., 2015*) were blocked with 5% normal goat serum for 1 hr to reduce non-specific reactions. They were then incubated with rabbit anti-cholecystokinin (*Hollander-Cohen et al., 2021*; *Himick and Peter, 1994*; *Sobrido-Cameán et al., 2020*; *Golan et al., 2016*; *Foucaud et al., 2008*; *Zhang et al., 2015b*; *Chu et al., 2015*; *Inoue et al., 2015*) (CCK-8) antibody (diluted 1:1000, Merck, C2581) for 16 hr at 4 °C. The same antibody had been previously used to mark CCK-positive cells in the gut of the red drum fish (*Khan et al., 2010*). Antibodies were diluted in PBS with 1% BSA and 0.3% Triton X-100. The slides were rinsed three times with PBS for 5 min and were incubated for 2 hr at room temperature with goat

anti-rabbit antibodies conjugated to Alexa674 fluorophore. HCR staining was performed according to the HCR RNA-FISH protocol for fresh-frozen or fixed-frozen tissue sections (*Choi et al., 2018*) (Molecular Instrument) on double-labeled transgenic fish *tg(LH:RFP, FSH:GFP)*. The detection stage was performed with probes against CCKBR(A)RNA (XM_017357750.2) and the amplification stage was performed using the 647 nm amplifier fluorophores. Sections were then counterstained with DAPI nuclear staining. After washing, slides were mounted with anti-fade solution (2% propyl gallate, 75% glycerol, in PBS) and imaged by confocal microscopy.

## In vivo assay for CCK and GnRH injections

Six-month-old zebrafish (five males and five females) were injected intraperitoneally with the following: (1) fish CCK peptide [(D-Y[SO3H]-L-G-W-M-D-F-NH (2), synthesized by GL Biochem] at a concentration of 10 ng/g or 100 ng/g body weight, 2) salmon GnRH analogue (D-Ala6,Pro9-Net)-mammalian GnRH (Bachem Inc, Torrance, CA) at a concentration of 100 ng/g body weight, (3) similar volumes of saline. Two hours post-injection, the fish were sedated using MS-222, bled from the heart as previously described (*Biran et al., 2012*), and decapitated. Pituitaries were dissected under a stereomicroscope and placed in Total RNA Isolation Reagent (Trizol). From each fish, between 15 μl and 20 μl of blood was collected. The blood was centrifuged at 970 × g 30 min and the plasma was separated and stored at −20 °C. LH and FSH expression in the pituitary was measured using real-time PCR (see *Figure 5—source data 1* for primer list). RNA extraction, reverse transcription of RNA, and real-time PCR were carried out as previously described (*Biran et al., 2008*). FSH secretion was measured in the plasma using ELISA for common carp, which was established in the Levavi Sivan lab using recombinant carp gonadotropins from in the yeast, and had been previously shown to be suitable for zebrafish (*Hollander-Cohen et al., 2018*).

## Generating the LOF mutants of the cck receptor

CCKBR(A) LOF was generated using CRISPR-Cas9 technology. Three single guide RNAs (sgRNA, *Figure 1—source data 1.*) were designed using the CHOPCHOP web tool (*Labun et al., 2019*) to specifically target coding regions in the CCKBR(A) gene (NCBI: XM_017357750.2;). Synthetic sgRNA (Sigma-Aldrich Israel Ltd) was co-injected with Cas9 into single-cell stage zebrafish embryos. Mature injected zebrafish were screened for gene mutation using high-resolution melt (HRM) curve analysis (*Segev-Hadar et al., 2021*) and bred with WT zebrafish to generate F1 heterozygous zebrafish. Out of the three designed sgRNA, guide number 2 was identified as the most efficient, creating the highest amount of mutated zebrafish. Mutated F1 heterozygous zebrafish were bred again to create the F2 generation containing a mix of genotypes: WT, heterozygous, and homozygous zebrafish.

Mixed genotype F2 siblings from the same spawning event were reared in the same tanks until sexual maturity was identified (5–6 mo). Tissues for H&E staining, RNA purification, and genotyping were collected from three groups of siblings (n=47).

For the genotyping of the mutation, fin clips were collected, and DNA was extracted using the HOTSHOT method (*Meeker et al., 2007*), amplified by PCR, and sequenced (sanger sequencing, Hylabs). Three types of mutations were identified and characterised for LOF: insertion of 12 nucleotides (CCKBR(A)$^{+12}$), insertion of 7 nucleotides (CCKBR(A)$^{+7}$), and depletion of one nucleotide (CCKBR(A)$^{-1}$;). LOF fish had contained one of the mutation types in each allele.

Pituitaries were collected for RNA purification and measured for LH and FSH expression using real-time PCR. RNA extraction, reverse transcription of RNA, and real-time PCR were carried out as previously described (*Biran et al., 2008*).

The abdomen of the zebrafish was fixed in 4% PFA and sent for H&E staining (Gavish Research Services (GRS)). 4 μm Slices of the abdomen containing the gonads were analyzed using FIJI (*Schindelin et al., 2012*). The gonad area and the different cell types in the gonad were identified according to the 'Histology atlas of the zebrafish' (van der ven, wester P 2003).

## Receptor signal transduction reporter assay

A receptor transactivation assay was performed according to *Biran et al., 2008* and *Mizrahi et al., 2019*. Briefly, the entire coding sequence of each CCK receptor variant was cloned from the DNA extracted from the different KO fish. Each coding sequence was inserted into pcDNA3.1 (Invitrogen). The constructs were transiently co-transfected with Cre-Luc reporter plasmid into COS7 cells (ATCC,

CRL-1651) Three μg of each construct was co-transfected with three μg of Cre-Luc. Forty-eight hours after transfection, cells were treated with tilapia CCK peptide (10 μM, eight dilutions of 1:3). The binding of the CCK peptide to the receptor elicits luciferase expression in a dose-dependent matter. The hormone treatment and the subsequent measurement of luciferase activities were carried out as previously described (*Biran et al., 2008*; *Mizrahi et al., 2019*). Non-linear curve fitting on baseline-corrected (control) values was performed using Prism version 10 software (GraphPad). COS7 cells underwent weekly measurements for mycoplasma contamination and morphology checks.

### Tissue distribution

Tissue distribution of Tilapia CCKAR, CCKBR, and CCKBR(A) mRNAs were determined by real-time PCR as previously described (*Biran et al., 2012*), Tissue samples were collected from three mature male tilapia and total RNA was extracted from gills, anterior brain, mid brain, hindbrain, pituitary, diencephalon, brain stem, hypothalamus, Liver, anterior intestine, posterior intestine, stomach, spleen, retina, testis, kidney and muscles. Total RNA and cDNA were prepared, and real-time PCR was performed as previously described (*Biran et al., 2012*). In short, cDNA (3 μL) was used as a template in a 20 μL reaction volume of platinum SYBR Green qPCR SuperMix (Invitrogen). Each set of primers was validated first for specificity and efficiency using six dilutions (5x) of sample cDNA. mRNA levels were normalized against the reference genes, EF1α, using the comparative threshold cycle method ($-\Delta CT$). Primer used for tilapia CCKAR, CCKBR, and CCKBR(A) described in *Figure 1—source data 1*.

### Starving experiment

Fish were acclimated for 2 wk and fed twice daily, a feed volume of 3% of their body weight. Six adult female zebrafish (0.43 ± 0.59 g) were randomly assigned to two experimental groups: the Fed group, which continued with the same feeding regimen, and the fasted group, which was not fed for 2 wk. After 2 wk, the brains of all zebrafish were dissected for RNA extraction. cDNA synthesis and RT-PCR were conducted following the methods previously described (*Biran et al., 2008*; *Hollander-Cohen et al., 2019*; *Mizrahi et al., 2019*). RT-PCR was conducted to measure ZF CCKb (XM_002665615.6) expression with EF1a used as the housekeeping gene.

### Statistical analysis

Statistical analysis was performed using Prism 9 software (GraphPad). Whiskers on bar plots represent mean ± SEM. In violin plots, the middle line represents the median, whereas the bottom and top lines represent the lower and upper quartiles, respectively. The datasets in all figures were tested for equal variances (using Bartlett's test) and normality (using D'Agostino and Pearson's test or Shapiro-Wilk test for smaller datasets). Dataset pairs that exhibited equal variances and normal distribution were compared using a two-tailed unpaired *t*-test (for two sets). For datasets with more than two sets, we used a one-way analysis of variance (ANOVA), followed by the Tukey-Kramer test. Datasets with different variances and/or non-Gaussian distributions were tested using two-tailed Mann-Whitney's test (for two sets) or Brown-Forsythe's one-way ANOVA, followed by Dunnett's T3 multiple comparisons test. To compare levels of secreted hormone in the ex vivo assay, the one-tailed paired *t*-test (Wilcoxon test) was performed, as the sample size was lower than 10. To compare the datasets of the in vivo assay of CCK and GnRH injections, and the gonadotrophs expression in the LOF fish, a nonparametric one-way ANOVA test (Kruskal-Wallis test) was used, as the datasets failed the Bartlett's test of equal variance. Significance was imparted at $p < 0.05$.

## Acknowledgements

The writers would like to acknowledge the contribution of Mr. Antony Pinot from the Mollard lab for its help in operating the two-photon microscopy, Einat Zelinger, and Daniel Waiger from the CSI Center for Scientific Imaging Faculty of Agriculture for, for their help in guidance in operating the confocal microscopy. The authors would like to thank Dr. Zohar Gavish at Gavish Research Services for performing the histological work. IPAM-BCM Platform, member of the national infrastructure France-BioImaging supported by the French National Research Agency (ANR-10-INBS-04). The Israel Science Foundation (ISF) support (grant number 1540/17). The U.S.-Israel Binational Science Foundation (Joint Funding Research Grants # NSF-BSF-1947541).

# Additional information

### Funding

| Funder | Grant reference number | Author |
|---|---|---|
| The French National Research Infrastructure for Biological Imaging | ANR-10-INBS-04 | Patrice Mollard |
| Israel Science Foundation | 1540/17 | Berta Levavi-Sivan |
| NSF-BSF | 1947541 | Berta Levavi-Sivan |

The funders had no role in study design, data collection and interpretation, or the decision to submit the work for publication.

### Author contributions

Lian Hollander-Cohen, Conceptualization, Data curation, Software, Investigation, Visualization, Methodology, Writing – original draft, Writing – review and editing; Omer Cohen, Miriam Shulman, Tomer Aizinkot, Investigation, Methodology; Pierre Fontanaud, Software; Omer Revah, Methodology; Patrice Mollard, Supervision, Methodology, Writing – review and editing; Matan Golan, Formal analysis, Supervision, Validation, Visualization, Methodology, Writing – original draft, Writing – review and editing; Berta Levavi-Sivan, Resources, Data curation, Supervision, Funding acquisition, Validation, Investigation, Writing – original draft, Writing – review and editing

### Author ORCIDs

Lian Hollander-Cohen ⓘ https://orcid.org/0000-0002-1590-5932
Patrice Mollard ⓘ https://orcid.org/0000-0002-2324-7589
Berta Levavi-Sivan ⓘ https://orcid.org/0000-0002-0183-9524

### Ethics

All experiments were approved by the Animal Welfare and Ethical Review Body of 486 Languedoc-Roussillon (APAFIS#745-2015060114396791) and by the Experimentation 487 Ethics Committee of the Hebrew University of Jerusalem (research number: AG-17-488 15126, Date: April 30, 2017).

Reviewer #2 (Public review): https://doi.org/10.7554/eLife.96344.3.sa1
Author response https://doi.org/10.7554/eLife.96344.3.sa2

---

# Additional files

### Supplementary files
• MDAR checklist

### Data availability

Gene expression data (figure 1A) was obtained from previously published dataset, deposited in GEO, GSE159470 https://doi.org/10.3390/ijms22126478.

The following previously published dataset was used:

| Author(s) | Year | Dataset title | Dataset URL | Database and Identifier |
|---|---|---|---|---|
| Hollander-Cohen L, Golan M, Levavi-Sivan B | 2021 | Differential regulation of gonadotropins as revealed by transcriptomes of distinct LH and FSH cells of fish pituitary | https://www.ncbi.nlm.nih.gov/geo/query/acc.cgi?acc=GSE159470 | NCBI Gene Expression Omnibus, GSE159470 |

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
