## [Editor Report · eLife Assessment]

This study presents **valuable** findings on the role of the satiety hormone cholecystokinin typically associated with feeding in the control of a pituitary hormone, FSH, which is a critical regulator of reproductive physiology. The authors provide **solid** pharmacological evidence that cholecystokinin is sufficient to regulate FSH and **compelling** genetic evidence that one of its receptors is required for gonadal development, with uncertainties remaining about the physiological regulation and necessity of the peptide. The work will be of interest to reproductive biologists, especially those with an interest in the endocrine control of fertility.

---

## [Referee Report · Reviewer #2 (Public review)]

Summary:

This manuscript builds on previous work suggesting that the CCK peptide is the releasing hormone for FSH in fishes, which is different than that observed in mammals where both LH and FSH release are under the control of GnRH. Based on data using calcium imaging as a readout for stimulation of the gonadotrophs, the researchers present data supporting the hypothesis that CCK stimulates FSH-containing cells in the pituitary. In contrast LH containing cells show a weak and variable response to CCK, but are highly responsive to GnRH. Data are presented that support the role of CCK in release of FSH. Researchers also state the functional overlap exists in the potency of GnRH to activate FSH cells, thus the two signalling pathways are not separate.

The results are of interest to the field because for many years the assumption has been that fishes use the same signalling mechanism. These data present an intriguing variation where a hormone involved in satiation acts in the control of reproduction.

Strengths:

The strengths of the manuscript are that researchers have shed light on different pathways controlling reproduction in fishes.

Weaknesses:

Weaknesses are that it is not clear if multiple ligand/receptors are involved (more than one CCK and more than one receptor?). The imaging of the CCK terminals and CCK receptors needs to be reinforced.

Comments on revisions:

The authors have responded to the comments with clarity and have made the important requested changes such as clarifying the CCK receptors (their expression and exactly which receptor was targeted), and emphasizing the interactions of CCK, namely that CCK induces LH secretion, but not to the same extent as FSH. All minor comments directed to the layout of the figures and text have been addressed. In summary, comments have been addressed satisfactorily.

---

## [Author Response]

The following is the authors’ response to the original reviews.

**Reviewer #1 (Public Review):**
Summary:The pituitary gonadotropins, FSH and LH, are critical regulators of reproduction. In mammals, synthesis and secretion of FSH and LH by gonadotrope cells are controlled by the hypothalamic peptide, GnRH. As FSH and LH are made in the same cells in mammals, variation in the nature of GnRH secretion is thought to contribute to the differential regulation of the two hormones. In contrast, in fish, FSH and LH are produced in distinct gonadotrope populations and may be less (or differently) dependent on GnRH than in mammals. In the present manuscript, the authors endeavored to determine whether FSH may be independently controlled by a distinct peptide, cholecystokinin (CCK), in zebrafish.Strengths:The authors demonstrated that the CCK receptor is enriched in FSH-producing relative to LH-producing gonadotropes, and that genetic deletion of the receptor leads to dramatic decreases in gonadotropin production and gonadal development in zebrafish. Also, using innovative in vivo and ex vivo calcium imaging approaches, they show that LH- and FSH-producing gonadotropes preferentially respond to GnRH and CCK, respectively. Exogenous CCK also preferentially stimulated FSH secretion ex vivo and in vivo.Weaknesses:The concept that there may be a distinct FSH-releasing hormone (FSHRH) has been debated for decades. As the authors suggest that CCK is the long-sought FSHRH (at least in fish), they must provide data that convincingly leads to such a conclusion. In my estimation, they have not yet met this burden. In particular, they show that CCK is sufficient to activate FSH-producing cells, but have not yet demonstrated its necessity. Their one attempt to do so was using fish in which they inactivated the CCK receptor using CRISPR-Cas9. While this manipulation led to a reduction in FSH, LH was affected to a similar extent. As a result, they have not shown that CCK is a selective regulator of FSH.

Our conclusion regarding the necessity of CCK signaling for FSH secretion is based on the following evidence:

(1) CCK-like receptors are expressed in the pituitary gland predominantly on FSH cells.

(2) Application of CCK to pituitaries elicits FSH cell activation and to a much lesser degree activation of LH cells. (calcium imaging assays)

(3) Application of CCK to pituitaries and by injections in-vivo significantly increased only FSH release.

(4) Mutating the FSH-specific CCK receptor in a different species of fish (medaka) also causes a complete shutdown of FSH production and phenocopies a fsh-mutant phenotype (Uehara, Nishiike et al. 2023).

Taken together, we believe that this data strongly supports the conclusion that CCK is necessary for FSH production and release from the fish pituitary. Admittedly, the overlapping effects of CCK on both FSH and LH cells in zebrafish (evident in both our calcium imaging experiments and especially in the KO phenotype) complicates the interpretation of the phenotype. We speculate that the effect of CCK on LH cells in zebrafish can be caused either by paracrine signaling within the gland or by the effects of CCK on GnRH neurons that were shown to express CCK receptors .

In the current version, we emphasize that CCK also induces LH secretion. Although it does not affect LH to the same extent as FSH, an overlap does exist. This is mentioned in the abstract and discussion.

Moreover, they do not yet demonstrate that the effects observed reflect the loss of the receptor's function in gonadotropes, as opposed to other cell types.

Although there is evidence for the expression of CCK receptor in other tissues, we do show a direct decrease of FSH and LH expression in the gonadotrophs of the pituitary of the mutant fish; taken together with its significant expression in FSH cells compared to the rest of the cells of the pituitary in the cell specific transcriptomic, it is the most reasonable explanation for the mutant phenotype.

Unfortunately, unlike in mice, technologies for conditional knockout of genes in specific cell types are not yet available for our model and cell types. Additional tissue distribution of the three receptors types of CCK was added in supplementary figure 1, from this tissue distribution it can be appreciated how in the pituitary only CCKBRA (our identified CCK receptor) is expressed, while in other tissues it is either not expressed or expressed with the additional CCK receptors that can compensate its activity.

It also is not clear whether the phenotypes of the fish reflect perturbations in pituitary development vs. a loss of CCK receptor function in the pituitary later in life. Ideally, the authors would attempt to block CCK signaling in adult fish that develop normally. For example, if CCK receptor antagonists are available, they could be used to treat fish and see whether and how this affects FSH vs. LH secretion.

While the observed gonadal phenotype of the KO (sex inversed fish) should have a developmental origin since it requires a long time to manifest, the effect of the KO on FSH and LH cells is probably more acute. Unfortunately a specific antagonist that affect only CCKRBA and not the other CCK receptors wasn’t identified yet.

In the Discussion, the authors suggest that CCK, as a satiety factor, may provide a link between metabolism and reproduction. This is an interesting idea, but it is not supported by the data presented. That is, none of the results shown link metabolic state to CCK regulation of FSH and fertility. Absent such data, the lengthy Discussion of the link is speculative and not fully merited.

In the revised manuscript, we provided data to link cck with metabolic status in supplementary figure 1 and modified the discussion to tone down the link between metabolic status to and reproductive state.

Also in the Discussion, the authors argue that "CCK directly controls FSH cells by innervating the pituitary gland and binding to specific receptors that are particularly abundant in FSH gonadotrophs." However, their imaging does not demonstrate innervation of FSH cells by CCK terminals (e.g., at the EM level).

Innervation of the fish pituitary does not imply a synaptic-like connection between axon terminals and endocrine cells. In fact, such connections are extremely rare, and their functionality is unclear. Instead, the mode of regulation between hypothalamic terminals and endocrine cells in the fish pituitary is more similar to "volume transmission" in the CNS, i.e. peptides are released into the tissue and carried to their endocrine cell targets by the circulation or via diffusion. A short explanation was added in lines 395-398 in the discussion

Moreover, they have not demonstrated the binding of CCK to these cells. Indeed, no CCK receptor protein data are shown.

Our revised manuscript includes detailed experiments showing the activation of the receptor by its homologous ligand, supplementary Figure 1 includes a transactivation assay of CCK to its receptor and the effect of the different mutants on the activation of the receptor. Unfortunately, no antibody is available against this fish specific receptor (one of the caveats of working with fish models); therefore, we cannot present receptor protein data.

The calcium responses of FSH cells to exogenous CCK certainly suggest the presence of functional CCK receptors therein; but, the nature of the preparations (with all pituitary cell types present) does not demonstrate that CCK is acting directly in these cells.

We agree with the reviewer that there are some disadvantages in choosing to work with a whole-tissue preparation. However, we believe that the advantages of working in a more physiological context far outweigh the drawbacks as it reflects the natural dynamics more precisely. Since our transcriptome data, as well as our ISH staining, show that the CCK receptor is exclusively expressed in FSH cells, it is improbable that the observed calcium response is mediated via a different pituitary cell type.

Indeed, the asynchrony in responses of individual FSH cells to CCK (Figure 4) suggests that not all cells may be activated in the same way. Contrast the response of LH cells to GnRH, where the onset of calcium signaling is similar across cells (Figure 3).

The difference between the synchronization levels of LH and FSH cells activity stems from the gap-junction mediated coupling between LH cells that does not exist between FSH cells(Golan, Martin et al. 2016). Therefore, the onset of calcium response in FSH cells is dependent on the irregular diffusion rate of the peptide within the preparation, whereas the tight homotypic coupling between LH cells generates a strong and synchronized calcium rise that propagates quickly throughout the entire population

The differences in connectivity between LH and FSH cells is mentioned in lines 194-195

Finally, as the authors note in the Discussion, the data presented do not enable them to conclude that the endogenous CCK regulating FSH (assuming it does) is from the brain as opposed to other sources (e.g., the gut).

We agree with the reviewer that, for now, we are unable to determine whether hypothalamic or peripheral CCK are the main drivers of FSH cells. While the strong innervation of the gland by CCK-secreting hypothalamic neurons strengthens the notion of a hypothalamic-releasing hormone and also fits with the dogma of the neural control of the pituitary gland in fish (Ball 1981), more experiments are required to resolve this question.

**Reviewer #2 (Public Review):**
Summary:This manuscript builds on previous work suggesting that the CCK peptide is the releasing hormone for FSH in fishes, which is different than that observed in mammals where both LH and FSH release are under the control of GnRH. Based on data using calcium imaging as a readout for stimulation of the gonadotrophs, the researchers present data supporting the hypothesis that CCK stimulates FSH-containing cells in the pituitary. In contrast, LH-containing cells show a weak and variable response to CCK but are highly responsive to GnRH. Data are presented that support the role of CCK in the release of FSH. Researchers also state that functional overlap exists in the potency of GnRH to activate FSH cells, thus the two signalling pathways are not separate. The results are of interest to the field because for many years the assumption has been that fishes use the same signalling mechanism. These data present an intriguing variation where a hormone involved in satiation acts in the control of reproduction.Strengths:The strengths of the manuscript are that researchers have shed light on different pathways controlling reproduction in fishes.Weaknesses:Weaknesses are that it is not clear if multiple ligand/receptors are involved (more than one CCK and more than one receptor?). The imaging of the CCK terminals and CCK receptors needs to be reinforced.Reviewer consultation summary:The data presented establish sufficiency, but not necessity of CCK in FSH regulation. The paper did not show that CCK endogenously regulates FSH in fish. This has not been established yet.

This is a very important comment, also raised by reviewer 1. To avoid repetition, please see our detailed response to the comment above.

The paper presents the pharmacological effects of CCK on ex vivo preparations but does not establish the in vivo physiological function of the peptide. The current evidence for a novel physiological regulatory mechanism is incomplete and would require further physiological experiments. These could include the use of a CCK receptor antagonist in adult fish to see the effects on FSH and LH release, the generation of a CCK knockout, or cell-specific genetic manipulations.

As detailed in the responses to the first reviewer, we cannot conduct conditional, cellspecific gene knockout in our model. However we did conducted KO and show the direct effect on FSH and LH secretion together with physiological characterisation of the mutant.

Zebrafish have two CCK ligands: ccka, cckb and also multiple receptors: cckar, cckbra and cckbrb. There is ambiguity about which CCK receptor and ligand are expressed and which gene was knocked out.

In the revised manuscript, we clarified which of the receptors are expressed (CCKRBA) and which receptor is targeted. We also provided data showing the specificity of the receptors (both WT and mutant) to the ligands. Supplementary 1 shows receptor cross-activation. The method also specifies the exact NCBI ID numbers of the targeted receptor and the antibody used for the immunostaining.

Blocking CCK action in fish (with receptor KO) affects FSH and LH. Therefore, the work did not demonstrate a selective role for CCK in FSH regulation in vivo and any claims to have discovered FSHRH need to be more conservative.

We agree with the reviewer that the overlap in the effect of CCK measured in the calcium activation of cells and in the KO model does not allow us to conclude selectivity. In this context, it is crucial to highlight that CCKRBA exhibits high expression on FSH cells but not on LH cells. Therefore, the effect of CCK on LH cells is likely paracrine or through GnRH neurons that were shown to express CCK receptors. In the current version, we emphasize that CCK also induces LH secretion. Although it does not affect LH to the same extent as FSH, an overlap does exist. This is mentioned in the abstract and discussion.

The labelling of the terminals with anti-CCK looks a lot like the background and the authors did not show a specificity control (e.g. anti-CCK antibody pre-absorbed with the peptide or anti-CCK in morphant/KO animals).

Figures colours had been updated to better visualise the specific staining of the antibody. Also, The same antibody had been previously used to mark CCK-positive cells in the gut of the red drum fish(Webb, Khan et al. 2010) , where a control (pre-absorbed with the peptide) experiment had been conducted.

**Recommendations for the authors:**

**Reviewer #1 (Recommendations For The Authors):**
Abstract:The authors have not yet established that CCK is the primary regulator of FSH in vivo.

In the new version, we highlight the leading effect of CCK on the reproductive axis, which includes FSH and LH.

Introduction:The authors need to make clear earlier in the Introduction that fish have two types of gonadotropes. This information comes too late (last paragraph) currently.

Added in line 42

They should discuss relevant data on the differential regulation of FSH and LH in fish, as a rationale for looking for different releasing factors.

This has been discussed in the first paragraph of the introduction

In the last sentence of the penultimate paragraph, the authors assume that it must be a hypothalamic factor that regulates FSH. Why is this necessarily the case? Are there data indicating that a hypothalamic factor is required for FSH production in fish?

This has been mentioned in the discussion, we do not deny that circulating CCK or CCK from other brain areas might affect FSH secretion in the pituitary (line 402-404). However, as the hypothalamus serves as the main gateway from the brain to the pituitary and contains hypophysiotropic CCK neurons it is the most reasonable assumption.

Results:In the first paragraph, the authors reference three types of CCK receptors, only one of which is expressed in the pituitary. The specific receptor should be named here.

The receptor name and NCBI id had been added in this paragraph.

Figure 1: What specificity controls were used for the ISH in Figure 1?

HCR- The method used to identify RNA expression and developed by Molecular Instruments (https://www.molecularinstruments.com/hcr-rnafish-protocols), do not require specific control as had been previously done with older ISH methods. The use of multiple short probes assure the specificity to the RNA.More over the expression is specific to the targeted cells.

In Figure 1D, the red square is missing in the KO fish (at low magnification).

This was fixed in the updated version.

In Figure 1G, the number of dots does not correspond to the number of animals described in the figure legend. Does each point represent an animal?

Each dot represent a fish. The order of the numbers in the legend didn’t match the order in the graph, this had been fixed in the last version

Figure 2A: It is not clear that all FSH (GFP) cells are double-labeled. Should all double-labeled cells appear white? Many appear as green. Some quantification of the proportion of co-labeling is needed. Also, the scale bars are too small to read. Perhaps add the size of the scale bars to the legend.

They are all double-labeled, as can be seen by the single-color images, since GFP fluorescence is stronger than RCaMP fluorescence, the double-labelling might be seen a green cells; a scale bar was added.

Figure 2C: Is the synchronous activity of LH cells here dependent on endogenous GnRH? Can these events be blocked with a GnRH receptor antagonist?

We currently do not have enough data to support this hypothesis and the in vivo 2 photon system is not optimal to answer these questions since these are spontaneous events which are difficult to predict. This is the main reason we moved to an ex vivo system. The similar response we receive when applying GnRH in the ex vivo system support it is GnRH activation.

Figure 4C: As some LH cells respond to CCK, can the authors really claim that CCK is a selective regulator of FSH? What explains the heterogeneity in the response of LH cells to CCK?

In this version, we highlight that CCK directly activates FSH but it is also affecting LH to some extent. However it is clear that the effect on FSH cells is more significant.

Figures 5A and B: With larger Ns, some of the trends might be significant (e.g., GnRH stimulated FSH release and CCK stimulated LH release).

Though there is a trend, the values in the Y axis reveal that the trend of response of FSH to GnRH and LH to CCK is lower then the distribution of the basal response (the before) in all of the graphs. Hence we do not believe a larger N will affect those results. We added the range of the secreted hormones concentrations in the result description to emphasize the difference in values,

Figures 5C and D: What explains the lack of an increase in LH secretion following GnRH treatment?

We did not measure LH Secretion in the plasma as we didn’t have enough blood, we do see an increase in LH transcription (see supplementary figure 5 – figure supplement 1)

Also, as mRNA levels were measured (in C), reference should be made to expression rather than transcription. Not all changes in mRNA levels reflect changes in transcription.Also, remove transcription from the legend. Reference to supplementary Figure 4 in the legend should be supplementary Figure 6. Finally, in C and D, distinguish males from females (as in 5A and B).

Modifications had been done according to the reviewer suggestions.

Figure legends:The figure legends are very long. One way to shorten them is to remove descriptions of the results. The legends should indicate what is in each figure, not the results of the experiments.

Modifications had been done according to the reviewer suggestions.

Sample sizes should be spelled out in the legends, as they are not in the M&M.

We made sure all sample sizes are mentioned in the legend

Materials and Methods:Section 1.1 can be removed as it repeats content presented elsewhere.

This section was removed

Section 1.5: It is unclear what this means: "blinding was not applied to ensure tractability" Please clarify.

This section was removed

**Reviewer #2 (Recommendations For The Authors):**
It appears that zebrafish have two ligands: ccka, cckb. Also multiple receptors: cckar, cckbra and cckbrb. Authors need to discuss this and clearly state which ligand and which receptor they are referring to in the manuscript.

We discussed the receptor type in the first paragraph of the results, the exact synthetic peptide used is described in the methods. The 8 amino acids of the mature CCK peptide are the same between CCKa and CCKb. A sentence regarding the specificity of the antibody to the mature CCK peptide was added in line 101.

"to GnRH puff application (300 μl of 30 μg/μl)"; (250 μl of 30 μg/ml CCK)Please give the final concentration to make it easy on the readers of the data.

The molarity of the final concentration was added.

(2.4) Differential calcium response underlies differential hormone. This section is a bit confusing to read, for example:"For that, we collected the medium perfused through our ex vivo system (Fig. 2a) and measured LH and FSH levels using a specific ELISA validated for zebrafish [31] while monitoring the calcium activity of the cells."So the authors did the ELISA while monitoring the activity (?). This sentence does not make sense: please rewrite it.

We modified this sentence in line 308-311

To functionally validate the importance of CCK signalling we used CRISPR-cas9 to generate loss-of-function (LOF) mutations in the pituitary- CCK receptor gene.The authors need to clearly state WHICH gene they inactivated: Zebrafish have three CCK-receptors, so "the pituitary receptor gene" needs to be defined.

Was added again in line 107, and is mentioned in the methods

Figure 3 is a crucial figure!Figure 3B: The data are not very convincing. Please state how thick the sections are in the figure legend (assuming these are adult pituitaries),

Added in the legend (figure 1C in the new version), slice thickness and adult fish.

Please show at least the merged image a high magnification view of the co-localization of the receptor with the cells.

This is figure 1 in the new revision, a magnified figure was added

Please give the scale bar size for 3B.

Scales for all images were added

Figure 3C: the co-localization of the terminals of the CCK and FSH cells shows very few cells expressing close to terminals.Important: Because the labelling of the terminals with anti-CCK looks a lot like the background, it is very important to show the control (anti-CCK antibody pre-absorbed with the peptide). The authors should have these data. The photo needs to have been taken at the same gain (contrast) and the photo showing the terminals.

This is a commercial antibody that had been previously validated for CCK in fish. The co-localization pattern resembles GnRH innervation in the pituitary. In fish when hypothalamic neurons innervate the pituitary they do not innervate all the cells, as this is an endocrine system, the peptide can travel to neighbouring cells via diffusion or aided blood flow (23). The images reveal the direct innervation of CCK in the pituitary and its proximity to FSH cells.

Figure 4c, on right. The text seems to be stretched as if the photo was adjusted without locking the aspect ratio. Please check the original images.

This has been fixed

Can the authors use different pseudo colours? Differentiating a double label of white versus yellow is very difficult, and thus the photo is not very convincing.

This had been changed to green and magenta

What is meant by "CCK-AB" antibody? Perhaps anti-CCK would be a better label

This has been fixed

Figure 5A: increase the magnification of the insets; the structure of the gonads is very difficult to see with clarity in these low mag images. The most obvious way to improve this figure is to reduce or eliminate the pie graph (not really necessary) and show a high magnification (and larger) image of the gonadal structure.

This is figure 1 in the new version, with magnification of the gonad next to each body section.

Discussion:" Moreover, in the zebrafish, as well as in other species, the functional overlap in gonadotropin signalling pathways is not limited to the pituitary but is also present in the gonad, through the promiscuity of the two gonadotropin receptors"The reasoning of this sentence is not clear: zebrafish do not use GnRH to control reproduction: they lack GnRH1 through genomic rearrangement (see Whitlock, Postlethwait and Ewer 2019) and KO of GnRH2/GnRH3 does not affect reproduction.

While GnRH KO model indicate a redundancy of GnRH in this axis in zebrafish, there is also ample evidence for its importance in regulating reproduction such as its effect on gonadotropin (Golan, Martin et al. 2016) and its use in spawning inductions in fish (Mizrahi and Levavi-Sivan 2023). We believe it is currently too soon to conclude that GnRH signalling is completely non relevant to reproduction in cyprinids.

**Reviewing Editor (Recommendations For The Authors):**
It would be interesting to see calcium imaging experiments in the CCKR receptor mutants to establish a more direct connection between peptide action and activity.

We added a receptor assay that reflect the non-activation of the mutated receptors by CCK (supplementary figure 1) , and compared it to the wild type that is activated. This show that: (1) CCK directly activate our identified receptor in FSH cells. (2) the mutated receptors are non-active.

"all homozygous fish (CCKR+12/+7/-1/ CCKR+12/+7/-1, n=12)"It may be better to write the genotype of fish separately as (CCKR+12/+12, CCKR+7/+7 and CCKR-1/-1, n=12) otherwise it seems as if all alleles occurred together in the same fish.

Modified according to the reviewer request

In Figure 1 scale bar legends are very small.

Description of the scale bars were added to the all the legends

Figure 1 legend "On the top right of each panel is the gender distribution" - fish have no gender but sex.

Modified according to the reviewer request

The authors should endeavour to improve the presentation of the figures. They should use a sans-serif font and check that text is not cut at the edge of figure panels, that scale bars are uniform and clearly labelled and fonts are of similar size and clearly legible. E.g. labels of the fish brain of Fig3A are very small.

We modified all the figures to adapt the font and the scales, we increased the size of the image in Figure 3a to make the labels clearer.

Please use the elife format to name supplementary figures, as Figure X - Figure Supplement Y (each supplement associated with one of the main figures).

Fixed

Peptide concentrations in the ex vivo experiments should also be given as molar concentrations not only as '250 μl of 30 μg/ml CCK'.

Fixed

"In contrast, FSH cells responded with a very low calcium rise in hormonal secretion in response to GnRH" - a very low rise in hormonal secretion

Fixed

Please clarify why you used a GnRH synthetic agonist and not the native peptide.

It is commonly used for spawning induction in fish (line 245); it has also been shown to directly affect the secretion of LH and FSH (Biran, Golan et al. 2014, Biran, Golan et al. 2014, Mizrahi, Gilon et al. 2019) , added to line 245.

References

Ball, J. (1981). "Hypothalamic control of the pars distalis in fishes, amphibians, and reptiles." General and comparative endocrinology 44(2): 135-170.

Biran, J., M. Golan, N. Mizrahi, S. Ogawa, I. S. Parhar and B. Levavi-Sivan (2014). "Direct regulation of gonadotropin release by neurokinin B in tilapia (Oreochromis niloticus)." Endocrinology 155(12): 4831-4842.

Biran, J., M. Golan, N. Mizrahi, S. Ogawa, I. S. Parhar and B. Levavi-Sivan (2014). "LPXRFa, the Piscine Ortholog of GnIH, and LPXRF Receptor Positively Regulate Gonadotropin Secretion in Tilapia (Oreochromis niloticus)." Endocrinology 155(11): 4391-4401.

Golan, M., A. O. Martin, P. Mollard and B. Levavi-Sivan (2016). "Anatomical and functional gonadotrope networks in the teleost pituitary." Scientific Reports 6: 23777.

Golan, M., E. Zelinger, Y. Zohar and B. Levavi-Sivan (2015). "Architecture of GnRH-Gonadotrope-Vasculature Reveals a Dual Mode of Gonadotropin Regulation in Fish." Endocrinology 156(11): 4163-4173.

Mizrahi, N., C. Gilon, I. Atre, S. Ogawa, I. S. Parhar and B. Levavi-Sivan (2019). "Deciphering Direct and Indirect Effects of Neurokinin B and GnRH in the Brain-Pituitary Axis of Tilapia." Front Endocrinol (Lausanne) 10: 469.

Mizrahi, N. and B. Levavi-Sivan (2023). "A novel agent for induced spawning using a combination of GnRH analog and an FDA-approved dopamine receptor antagonist." Aquaculture 565: 739095.

Uehara, S. K., Y. Nishiike, K. Maeda, T. Karigo, S. Kuraku, K. Okubo and S. Kanda (2023). "Cholecystokinin is the follicle-stimulating hormone (FSH)-releasing hormone." bioRxiv: 2023.2005.2026.542428.

Webb, K. A., Jr., I. A. Khan, B. S. Nunez, I. Rønnestad and G. J. Holt (2010). "Cholecystokinin: molecular cloning and immunohistochemical localization in the gastrointestinal tract of larval red drum, Sciaenops ocellatus (L.)." Gen Comp Endocrinol 166(1): 152-159.